# Analysis of the mechanosensor channel functionality of TACAN

**Yiming Niu[1], Xiao Tao[1,2], George Vaisey[1,2], Paul Dominic B Olinares[3], Hanan Alwaseem[4], Brian T Chait[3], Roderick MacKinnon[1,2]***

[1]Laboratory of Molecular Neurobiology and Biophysics, Rockefeller University, New York, United States; [2]Howard Hughes Medical Institute, New York, United States; [3]Laboratory of Mass Spectrometry and Gaseous Ion Chemistry, Rockefeller University, New York, United States; [4]Proteomics Resource Center, Rockefeller University, New York, United States

**Abstract** Mechanosensitive ion channels mediate transmembrane ion currents activated by mechanical forces. A mechanosensitive ion channel called TACAN was recently reported. We began to study TACAN with the intent to understand how it senses mechanical forces and functions as an ion channel. Using cellular patch-recording methods, we failed to identify mechanosensitive ion channel activity. Using membrane reconstitution methods, we found that TACAN, at high protein concentrations, produces heterogeneous conduction levels that are not mechanosensitive and are most consistent with disruptions of the lipid bilayer. We determined the structure of TACAN using single-particle cryo-electron microscopy and observed that it is a symmetrical dimeric transmembrane protein. Each protomer contains an intracellular-facing cleft with a coenzyme A cofactor, confirmed by mass spectrometry. The TACAN protomer is related in three-dimensional structure to a fatty acid elongase, ELOVL7. Whilst its physiological function remains unclear, we anticipate that TACAN is not a mechanosensitive ion channel.

**\*For correspondence:**
mackinn@mail.rockefeller.edu

**Competing interest:** The authors declare that no competing interests exist.

## Introduction

Mechanosensitive ion channels (MSCs) open in response to mechanical forces (*Guharay and Sachs, 1984*; *Guharay and Sachs, 1985*; *Kung, 2005*; *Sachs, 2010*). When the channels open, ions flow across the cell membrane, triggering subsequent biochemical processes that ultimately represent a cellular response to the applied mechanical force. This coupling of transmembrane (TM) ion flow to mechanical forces underlies some forms of osmoregulation, cell and organ growth, blood pressure regulation, touch, and hearing (*Chalfie, 2009*; *Coste et al., 2010*; *Pan et al., 2013*; *Peyronnet et al., 2012*; *Woo et al., 2015*). Several MSCs have been discovered and characterized (*Kefauver et al., 2020*). Recently, a new MSC in mammals called TACAN was reported and proposed to mediate mechanical pain (*Beaulieu-Laroche et al., 2020*). TACAN, originally identified in a proteomics screen and called TMEM120A, was categorized as a nuclear envelope protein (NET29) that participates in lipid metabolism (*Batrakou et al., 2015*; *Byerly et al., 2010*; *Haakonsson et al., 2013*; *Lee et al., 2005*; *Rosell et al., 2014*). Adipocyte-specific TMEM120A knockout mice exhibited a lipodystrophy syndrome similar to human familial partial lipodystrophy FPLD2 (*Czapiewski et al., 2021*).

As our laboratory studies the biophysical mechanisms by which MSCs transduce mechanical forces and conduct ions across membranes, we were intrigued by TACAN's potential role as an MSC and set out to examine this function and report our findings here.

## Results

### Functional analysis in cells and reconstituted membranes

We sought to reproduce the mechanically evoked currents reported when TACAN is expressed in cells (*Beaulieu-Laroche et al., 2020*). Using CHO cells, similar to those used in the original study, we did not observe pressure-evoked currents in excised membrane patches (*Figure 1A,B*). Background channels that were not sensitive to the pressure steps were observed in CHO cells expressing either TACAN or the M2 muscarinic receptor as a control. Similarly, TACAN expressed in a Piezo1 knockout HEK cell line did not elicit pressure-activated channels (*Figure 1C,D*). Purified TACAN protein reconstituted into giant unilamellar vesicles (GUVs) of soy L-α-phosphatidylcholine (soy-PC) also did not yield pressure-activated channels in membrane patches isolated from the GUVs (*Figure 1E*). We note that previously we have successfully recorded mechanosensitive TRAAK channels in GUVs using the identical approach (*Brohawn et al., 2014*).

When TACAN was expressed, purified, and reconstituted into both GUVs and planar lipid bilayers at high protein-to-lipid ratios (≥1:100, m:m), transient currents were observed, as shown in *Figure 2*.

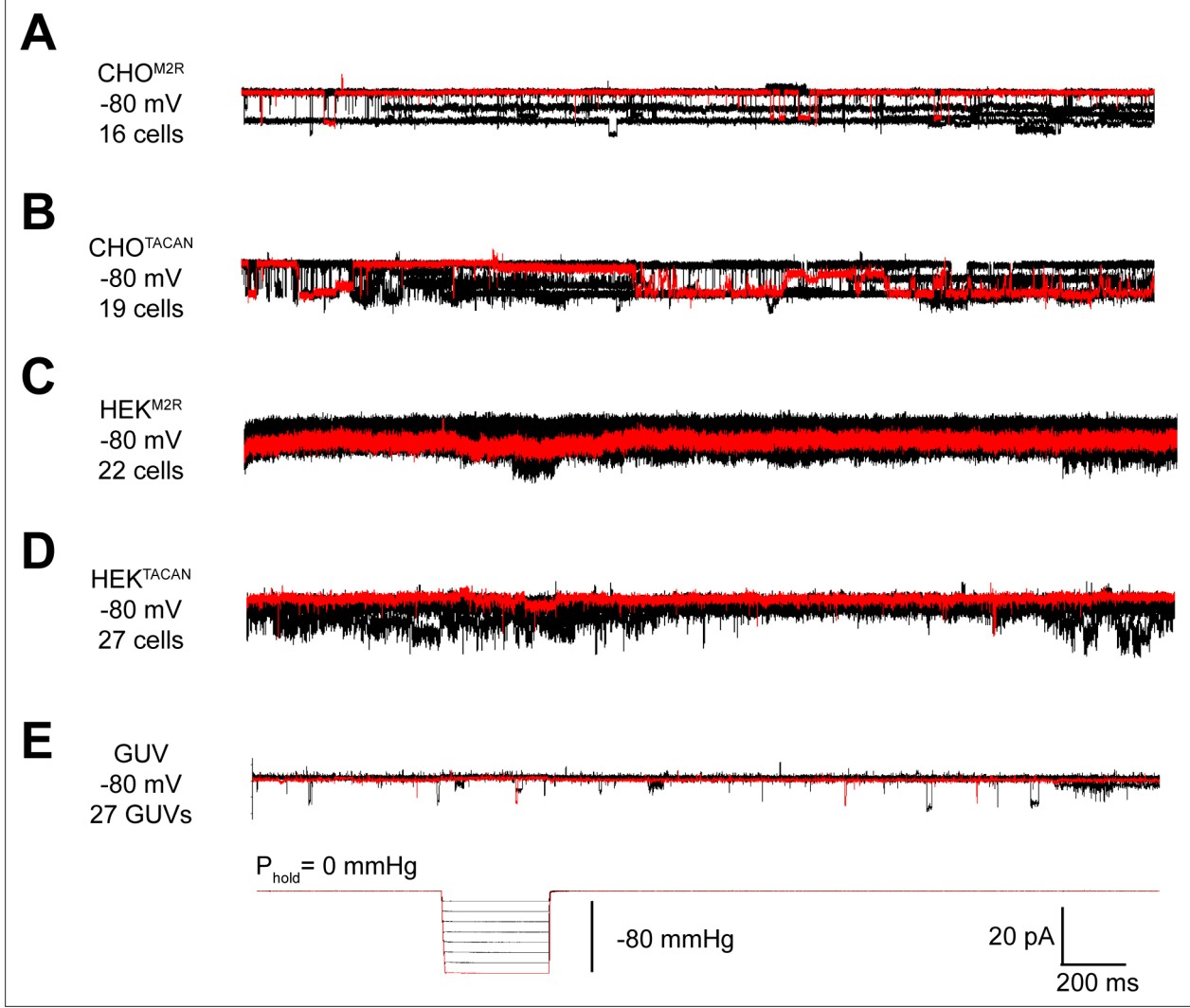

**Figure 1.** TACAN does not produce mechanically evoked currents. (**A, B**) Representative excised inside-out patch recordings of M2 muscarinic receptor (M2R, **A**) and TACAN (**B**) transfected into CHO-K1 cells. (**C, D**) Representative excised inside-out patch recordings of M2R (**C**) and TACAN (**D**) transfected into piezo-1 knockout HEK-293T cells. (**E**) Representative excised inside-out patch recording of TACAN reconstituted in giant unilamellar vesicles (GUVs). All recordings were performed with identical pipette and bath solution containing 10 mM HEPES pH 7.4, 140 mM KCl, and 1 mM MgCl$_2$ (~300 Osm/L). Traces were obtained holding at –80 mV with a pressure pulse protocol shown at the bottom: 0 to –80 mmHg with 10 mmHg step. Traces colored in red represent the observed currents with –80 mmHg pressure pulse.

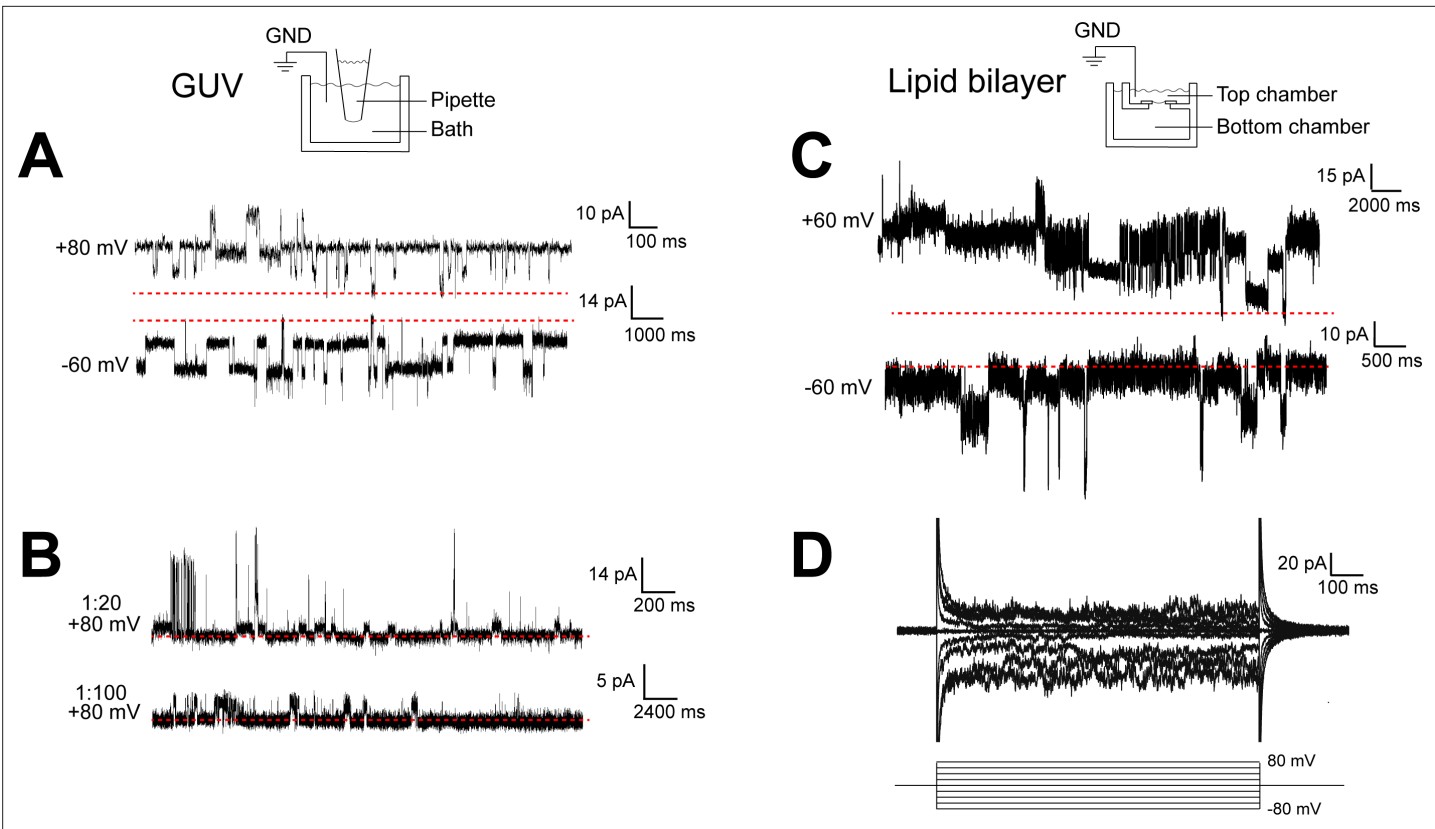

**Figure 2.** TACAN produces heterogenous currents in reconstituted systems. (**A, B**) Representative recordings of TACAN from excised giant unilamellar vesicle (GUV) patches. Symmetrical buffers (10 mM HEPES pH 7.4, 140 mM KCl, 1 mM MgCl$_2$) were used in pipette and bath. The dashed red lines indicate the baseline currents. (**A**) Traces from GUVs at 1:20 protein-to-lipid ratio (w/w) holding at +80 mV and –60 mV. (**B**) Traces from GUVs at 1:20 and 1:100 protein-to-lipid ratio (w/w) holding at +80 mV. (**C, D**) Representative traces of TACAN reconstituted in a lipid bilayer. Symmetrical buffers (10 mM HEPES pH 7.4, 150 mM KCl) were used in top and bottom chambers. The dashed red lines indicate the baseline currents. (**C**) Traces while holding at +60 mV and –60 mV. (**D**) Traces recorded during a voltage family from –80 to +80 mV in 20 mV increment.

These currents were insensitive to pressure applied to patches isolated from the GUVs (*Figure 1E*) and heterogeneous in amplitude (*Figure 2A–C*). These properties do not resemble aspects of currents from known ion channels but might suggest that TACAN renders the membrane transiently leaky when reconstituted at high protein concentrations.

## Structural analysis of TACAN

Alongside the functional characterization, we analyzed the structure of TACAN determined at 3.5 Å resolution using single-particle cryo-EM. Details of the structure determination are given in Materials and methods and *Table 1* (*Figure 3—figure supplements 1 and 2*). TACAN is an α-helical TM protein that forms a symmetric dimer (*Figure 3A*). The orientation of the protein with respect to the cytoplasm is unknown; however, the charge distribution on TACAN (*von Heijne, 1986*) as well as the possible presence of an enzyme active site exposed to the cytoplasm (discussed below) suggests the orientation shown (*Figure 3B*). Each protomer consists of six TM helices (S1–S6), which form a barrel surrounding a tunnel open to the cytoplasm (*Figure 3C*). Non-continuous density was observed inside the tunnel, suggesting the presence of a small, non-protein molecule (*Figure 3—figure supplement 3A*). The two protomers of the TACAN dimer bury an extensive surface area of 3049 Å$^2$, mediated through the TM domain as well as two long N-terminal helices that form a coiled coil (*Figure 3A*, *Figure 3—figure supplement 3B*).

The DALI three-dimensional structure comparison server (*Holm and Rosenström, 2010*) identified a homologous protein called ELOVL7, a long-chain fatty acid (FA) elongase (*Figure 4*; *Nie et al., 2021*). This enzyme catalyzes the first step in the FA elongation cycle by transferring an acetyl group from malonyl-CoA onto long-chain FA-CoA (*Naganuma et al., 2011*). As shown in *Figure 4A*, the

**Table 1.** Cryo-EM data collection and refinement statistics, related to *Figures 3 and 4*.

| | TACAN[WT] | TACAN[H196A H197A] |
|---|---|---|
| EMDB ID | EMD-24107 | EMD-24108 |
| PDB ID | 7N0K | 7N0L |
| *Data collection* | | |
| Microscope | Titan Krios | |
| Detector | K2 summit | K3 summit |
| Voltage (kV) | 300 | 300 |
| Pixel size (Å) | 1.03 | 0.515 |
| Total electron exposure (e⁻/Å²) | 75.4 | 56.6 |
| Defocus range (μm) | 0.7–2.1 | 0.8–2.2 |
| Micrographs collected | 2,071 | 10,541 |
| *Reconstruction* | | |
| Final particle images | 110,090 | 155,946 |
| Pixel size (Å) | 1.03 | 1.03 |
| Box size (pixels) | 256 | 256 |
| Resolution (Å) (FSC = 0.143) | 3.5 | 2.8 |
| Map sharpening B-factor (Å²) | –20 | –3.4 |
| *Model composition* | | |
| Non-hydrogen atoms | 5,156 | 5,272 |
| Protein residues | 626 | 626 |
| Ligands | 0 | 2 |
| Metals | 0 | 0 |
| *Refinement* | | |
| Model-to-map CC (mask) | 0.77 | 0.80 |
| Model-to-map CC (volume) | 0.73 | 0.81 |
| R.m.s deviations | | |
| Bond length (Å) | 0.003 | 0.003 |
| Bond angles (°) | 0.54 | 0.52 |
| *Validation* | | |
| MolProbity score | 2.09 | 2.22 |
| Clash score | 7.86 | 9.10 |
| *Ramachandran plot* | | |
| Outliers (%) | 0 | 0 |
| Allowed (%) | 0.98 | 1.95 |
| Favored (%) | 99.02 | 98.05 |
| Rotamer outliers (%) | 7.46 | 9.23 |
| C-beta deviations (%) | 0 | 0 |

FSC: Fourier shell correlation.

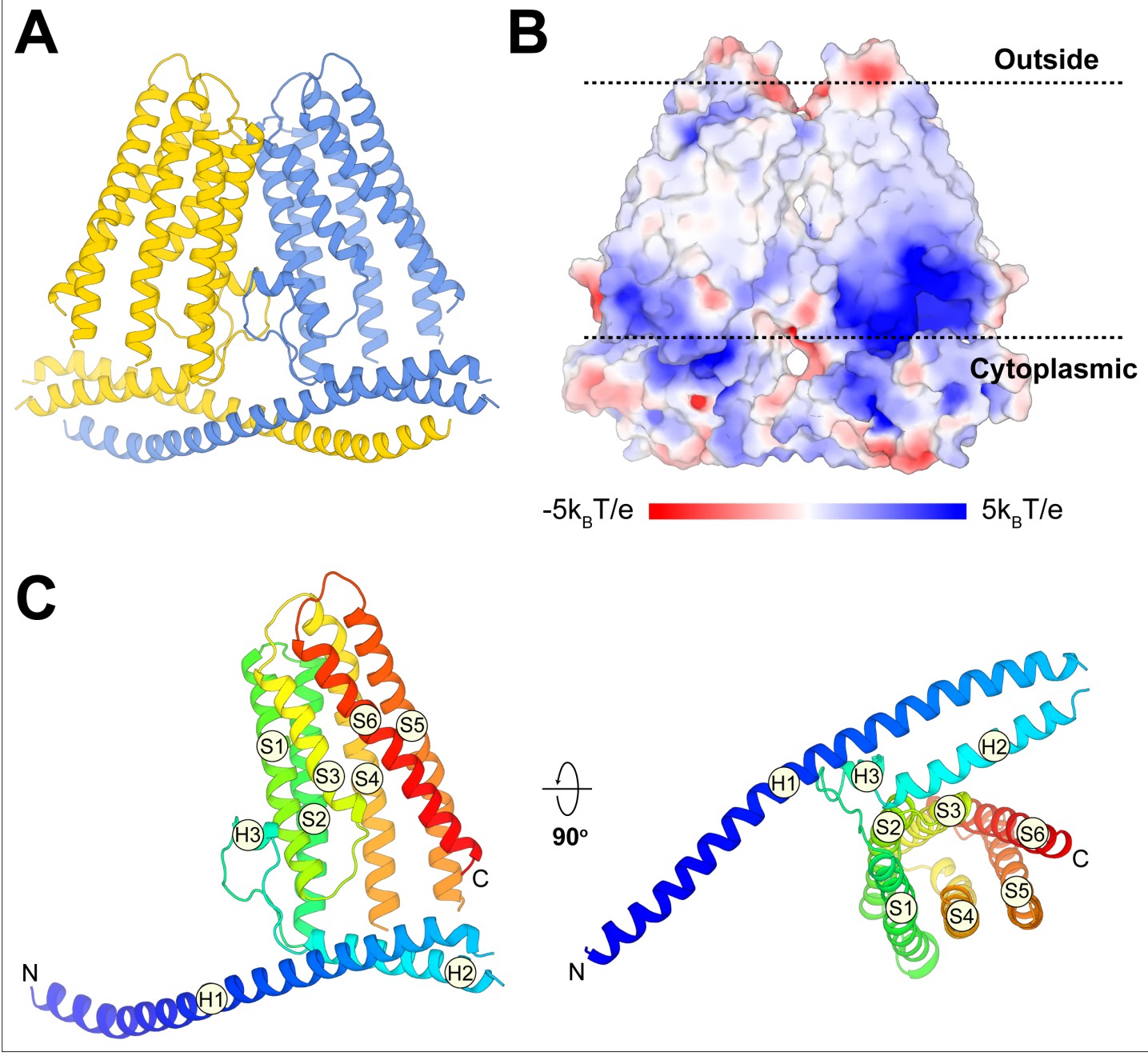

**Figure 3.** Overall structure of TACAN. (**A**) Cartoon representation of the TACAN dimer with each protomer colored uniquely. (**B**) Surface charge distribution and the possible orientation of TACAN, blue and red representing the positive and negative charges, respectively. The membrane is demarcated by dashed lines. (**C**) Tertiary structure of TACAN protomer viewed from the side and the cytoplasmic side. The protein is colored rainbow from N-terminus (blue) to C-terminus (red). The six transmembrane helices (S1–S6), two horizontal helices (H1 and H2), as well as a short helix (H3) in between are labeled.

The online version of this article includes the following figure supplement(s) for figure 3:

**Figure supplement 1.** Cryo-EM analysis of wild-type TACAN.

**Figure supplement 2.** Representative density in the cryo-EM map of wild-type TACAN.

**Figure supplement 3.** Structural analysis of wild-type TACAN.

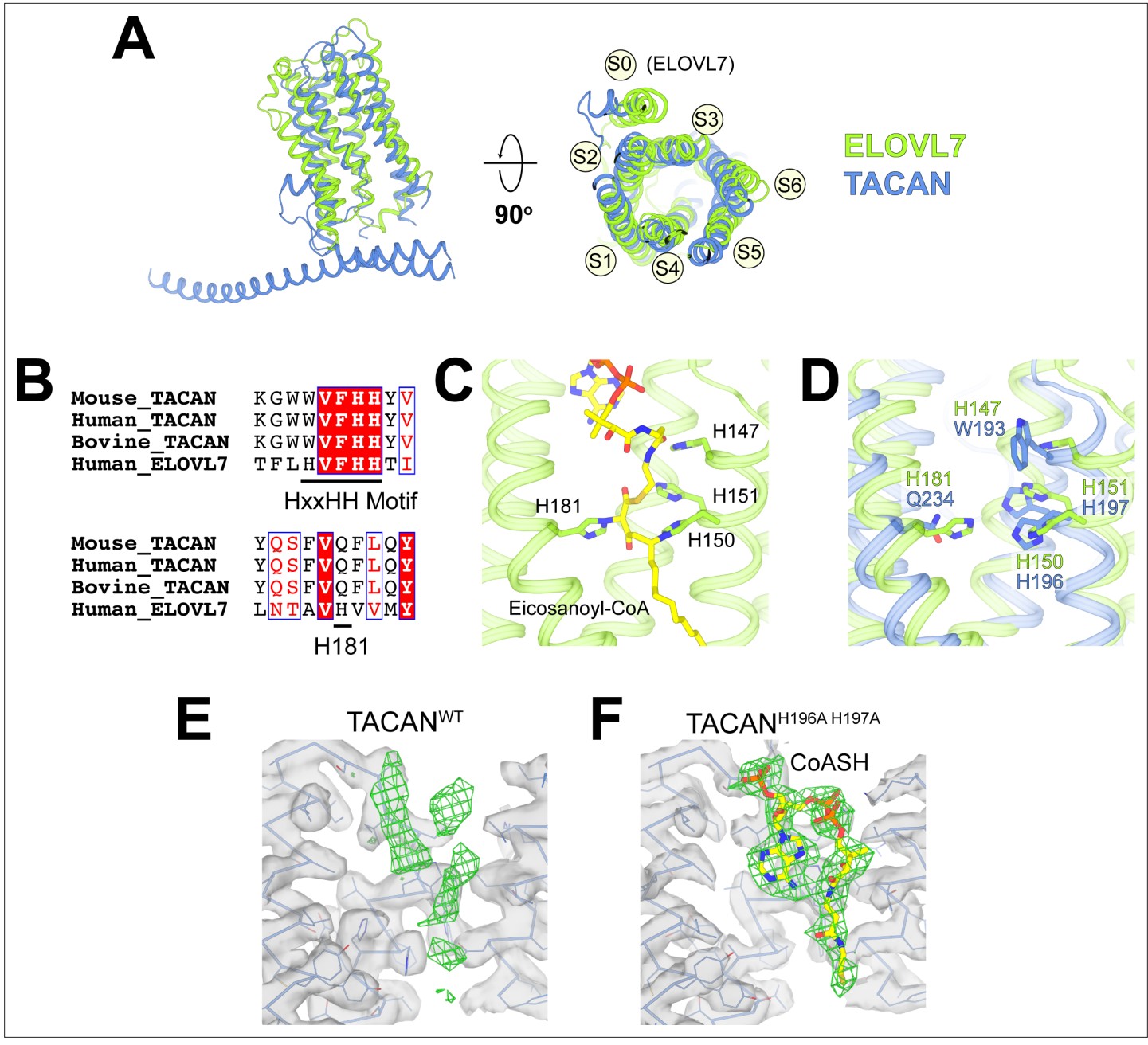

**Figure 4.** TACAN shares structural homology to the fatty acid elongase ELOVL7. (**A**) Superposition between TACAN (blue) and ELOVL7 (green) protomers. Transmembrane helices are labeled to correspond with the topology of TACAN. The extra transmembrane helix in ELOVL7 is labeled as S0. (**B**) Sequence alignment of TACAN from different species and human ELOVL7 with conserved residues highlighted. The catalytically important HxxHH motif (His147, His150, and His151) and His181 in ELOVL7 are underlined. (**C**) Structure details of the interactions between the HxxHH motif, His181 (sidechains shown as sticks), and eicosanoyl-CoA (shown as sticks) in ELOVL7 (PDB: 6Y7F). His150 and His181 are covalently linked to eicosanoyl-CoA. (**D**) Zoom-in view of the ELOVL7 (green) catalytic center with TACAN (blue) superimposed. (**E, F**) The non-protein density (green mesh) in the narrow tunnel of wild-type (**E**) and His196Ala, His197Ala mutant of TACAN (**F**). Protein density is represented as transparent surface (gray) with protein shown as lines and ribbons. The two maps are shown at the same contour level. CoASH in mutant TACAN is shown as sticks and colored according to atom type.

The online version of this article includes the following figure supplement(s) for figure 4:

**Figure supplement 1.** Cryo-EM analysis of the His196Ala His197Ala mutant TACAN.

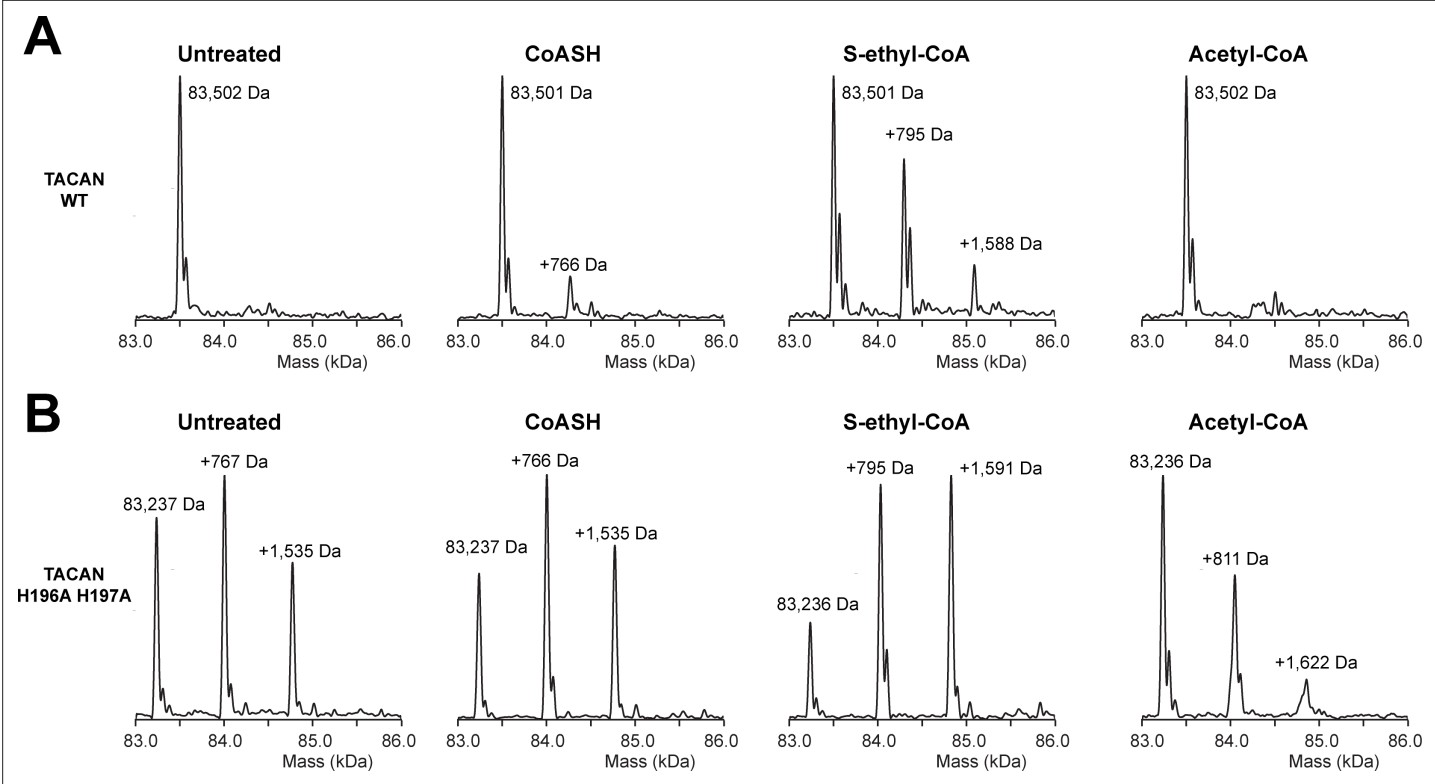

**Figure 5.** Native mass spectrometry indicates the presence of coenzyme A in the mutant TACAN sample. (**A, B**) Mass species detected in purified wild-type (**A**) and His196Ala, His197Ala mutant (**B**) TACAN protein without treatment ("untreated"), or incubated with CoASH (MW = 767.5 Da), S-ethyl-CoA (MW = 795.6 Da), or acetyl-CoA (MW = 809.6 Da).

The online version of this article includes the following figure supplement(s) for figure 5:

**Figure supplement 1.** TACAN is co-purified with coenzyme A molecules.

TM domain in TACAN is indeed similar to ELOVL7. The tunnel in ELOVL7 is lined by catalytically important histidine residues and contains a covalently linked eicosanoyl-CoA molecule (*Figure 4B,C*). TACAN conserves two of the four histidine residues (*Figure 4B,D*). To determine the identity of the small molecule implied by the broken density in the tunnel of TACAN (*Figure 4E* and *Figure 3—figure supplement 3A*), we determined the structure of TACAN with His196 and His197 mutated to alanine at 2.8 Å resolution (*Table 1*, *Figure 4—figure supplement 1*). Our rationale was that if the His residues are catalytically important – by analogy to ELOVL7 – then their mutation might influence the occupancy of a potential cofactor. The map showed clearer density consistent with a coenzyme A molecule (CoASH) (*Figure 4F*). Native mass spectrometry (nMS) was used to confirm the identity as CoASH (*Figure 5*). As shown in *Figure 5B*, the purified TACAN[H196A H197A] sample contains a mixture of the 83,237 Da, +767 Da, and +1535 Da mass species, corresponding to an apo form, one and two CoASH bound forms, respectively. After incubation with CoASH, some fraction of the apo form shifts to one and two CoASH bound forms. Additionally, the +767 Da and +1535 Da species are replaced by +795 Da and +1591 Da or +811 Da and +1622 Da species after incubation with the two CoASH analogues S-ethyl-CoA or Acetyl-CoA, corresponding to the CoASH analogue bound forms. In the purified TACAN[WT] sample, the apo form is dominant and incubation with S-ethyl-CoA shifts it to one and two analogue bound forms (*Figure 5A*). Together, these data are consistent with our cryo-EM results and indicate that TACAN is co-purified with endogenous coenzyme A.

It is noteworthy that CoASH binds with different conformations in TACAN from ELOVL7 (*Figure 5—figure supplement 1A,B*; *Nie et al., 2021*). In addition, no enzymatic activity was observed for TACAN using a free-CoA detection assay (for details, see Materials and methods), which demonstrated robust activity for ELOVL7 (*Figure 5—figure supplement 1C*), thus TACAN does not appear to catalyze the same reaction as ELOVL7. If TACAN is a coenzyme A-dependent enzyme, its substrate is unknown.

## Discussion

We undertook this study to understand how TACAN functions as an MSC, but have been unable to replicate evidence of MSC activity. We observe no channel activity in the plasma membrane of cells expressing TACAN and the heterogeneous-in-amplitude currents (without mechanosensitive properties) that we observe when we reconstitute TACAN at high protein concentrations are not consistent with other native biological channels that we have studied.

Structurally, TACAN is related to coenzyme A-dependent FA elongases; however, without further data we cannot conclude that TACAN itself functions as an enzyme. It also remains to be determined which membranes in a cell express TACAN.

In conclusion, we do not find evidence that TACAN is a mechanosensitive ion channel. The strength of this conclusion is in the electrophysiological interrogation. The structure, because it looks like a known enzyme, is compatible with the 'not a channel' conclusion, but the structure alone would not make a strong argument. A number of ion channels, including CLC channels (*Dutzler et al., 2002*; *Dutzler et al., 2003*; *Feng et al., 2012*; *Park et al., 2017*; *Park and MacKinnon, 2018*), TMEM16 (*Dang et al., 2017*; *Paulino et al., 2017*), and CFTR (*Liu et al., 2017*; *Zhang et al., 2017*; *Zhang et al., 2018*), are not obviously ion channels based on their structures and indeed each are fairly indistinguishable from proteins exhibiting non-channel functions.

# Materials and methods

**Key resources table**

| Reagent type (species) or resource | Designation | Source or reference | Identifiers | Additional information |
|---|---|---|---|---|
| Gene (*Mus musculus* TMEM120A) | *M. musculus* TACAN | Synthetic | | Synthesized at GeneWiz. |
| Gene (*Homo sapiens* TMEM120A) | *H. sapiens* TACAN | Synthetic | | Synthesized at GeneWiz. |
| Gene (*Homo sapiens* ELOVL7) | *H. sapiens* ELOVL7 | Synthetic | | Synthesized at GeneWiz. |
| Strain, strain background (*Escherichia coli*) | DH10Bac | Thermo Fisher Scientific | 10361012 | |
| Recombinant DNA reagent | TACAN-eGFP BacMam | This study | | |
| Recombinant DNA reagent | ELOVL7-eGFP BacMam | This study | | |
| Recombinant DNA reagent | Halo-M2R-eGFP BacMam | This study | | |
| Cell line (*Spodoptera frugiperda*) | Sf9 | ATCC | CRL-1711 | Cells purchased from ATCC, and we have confirmed there is no mycoplasma contamination |
| Cell line (Chinese hamster) | CHO-K1 | ATCC | CRL-9618 | Cells purchased from ATCC, and we have confirmed there is no mycoplasma contamination |
| Cell line (*Homo sapiens*) | HEK293S GnTI⁻ | ATCC | CRL-3022 | Cells purchased from ATCC, and we have confirmed there is no mycoplasma contamination |

*Continued on next page*

*Continued*

| Reagent type (species) or resource | Designation | Source or reference | Identifiers | Additional information |
|---|---|---|---|---|
| Cell line (*Homo sapiens*) | Piezo1 knockout HEK293T | https://digitalcommons.rockefeller.edu/cgi/viewcontent.cgi?article=1422&context=student_theses_and_dissertations | | We have confirmed there is no mycoplasma contamination |
| Chemical compound, drug | SF-900 II SFM medium | Gibco | 11330-032 | |
| Chemical compound, drug | L-Glutamine (100 ×) | Gibco | 25030-081 | |
| Chemical compound, drug | Pen Strep | Gibco | 15140-122 | |
| Chemical compound, drug | Grace's insect medium | Gibco | 11605-094 | |
| Chemical compound, drug | Freestyle 293 medium | Gibco | 12338-018 | |
| Chemical compound, drug | DMEM/F-12 medium | Gibco | 11605-094 | |
| Chemical compound, drug | DMEM | Gibco | 11965-118 | |
| Chemical compound, drug | Fetal bovine serum | Gibco | 16000-044 | |
| Chemical compound, drug | Cellfectin II reagent | Invitrogen | 10362100 | |
| Chemical compound, drug | FuGENE HD transfection reagent | Promega | E2312 | |
| Chemical compound, drug | Cholesteryl hemisuccinate (CHS) | Anatrace | CH210 | |
| Chemical compound, drug | n-Decyl-β-D-maltopyranoside (DM) | Anatrace | D322S | |
| Chemical compound, drug | Lauryl maltose neopentyl glycol (LMNG) | Anatrace | NG310 | |
| Chemical compound, drug | Digitonin | Millipore Sigma | 300410 | |
| Chemical compound, drug | Coenzyme A trilithium salt (CoASH) | Sigma-Aldrich | C3019 | |
| Chemical compound, drug | Acetyl coenzyme A sodium salt (acetyl-CoA) | Sigma-Aldrich | A2056 | |
| Chemical compound, drug | S-Ethyl-coenzyme A sodium salt (S-ethyl-CoA) | Jena-Biosciences | NU-1168 | |
| Chemical compound, drug | Malonyl coenzyme A lithium salt (malonyl-CoA) | Sigma-Aldrich | M4263 | |
| Chemical compound, drug | Stearoyl coenzyme A lithium salt (stearoyl-CoA) | Sigma-Aldrich | S0802 | |
| Chemical compound, drug | (1 H, 1 H, 2 H, 2H-Perfluorooctyl) phosphocholine (FFC8) | Anatrace | F300F | |
| Commercial assay or kit | CNBr-activated Sepharose beads | GE Healthcare | 17-0430-01 | |
| Commercial assay or kit | Superdex 200 Increase 10/300 GL | GE Healthcare Life Sciences | 28990944 | |

*Continued on next page*

*Continued*

| Reagent type (species) or resource | Designation | Source or reference | Identifiers | Additional information |
|---|---|---|---|---|
| Commercial assay or kit | R1.2/1.3 400 mesh Au holey carbon grids | Quantifoil | 1210627 | |
| Commercial assay or kit | Coenzyme A (CoA) Assay Kit | Sigma-Aldrich | MAK034 | |
| Software, algorithm | RELION 3.0 | https://doi.org/10.7554/eLife.42166.001 | http://www2.mrc-lmb.cam.ac.uk/relion | |
| Software, algorithm | RELION 3.1 | https://doi.org/10.1101/798066 | http://www2.mrc-lmb.cam.ac.uk/relion | |
| Software, algorithm | MotionCor2 | https://doi.org/10.1038/nmeth.4193 | http://msg.ucsf.edu/em/software/motioncor2.html | |
| Software, algorithm | Gctf 1.0.6 | https://doi.org/10.1016/j.jsb.2015.11.003 | https://www.mrc-lmb.cam.ac.uk/kzhang/Gctf/ | |
| Software, algorithm | CtfFind4.1.8 | https://doi.org/10.1016/j.jsb.2015.08.008 | http://grigorieflab.janelia.org/ctffind4 | |
| Software, algorithm | CryoSPARC 2.9.0 | https://doi.org/10.7554/eLife.46057.001 | https://cryosparc.com/ | |
| Software, algorithm | COOT | https://doi.org/10.1107/S0907444910007493 | http://www2.mrc-lmb.cam.ac.uk/personal/pemsley/coot | |
| Software, algorithm | PHENIX | https://doi.org/10.1107/S0907444909052925 | https://www.phenix-online.org | |
| Software, algorithm | Adobe Photoshop version 16.0.0 (for figure preparation) | Adobe Systems, Inc. | | |
| Software, algorithm | GraphPad Prism version 8.0 | GraphPad Software | | |
| Software, algorithm | MacPyMOL: PyMOL v2.0 Enhanced for Mac OS X | Schrodinger LLC | https://pymol.org/edu/?q=educational/ | |
| Software, algorithm | Chimera | https://doi.org/10.1002/jcc.20084 | https://www.cgl.ucsf.edu/chimera/download.html | |
| Software, algorithm | Serial EM | https://doi.org/10.1016/j.jsb.2005.07.007 | http://bio3d.colorado.edu/SerialEM | |
| Software, algorithm | pClamp | Axon Instruments, Inc | | |
| Software, algorithm | Thermo Xcalibur Qual Browser (v. 4.2.47) | Thermo Fisher Scientific | | |
| Software, algorithm | UniDec v. 4.2.0 | *Marty et al., 2015*; *Reid et al., 2018* | https://github.com/michaelmarty/UniDec/releases | |

## Protein expression and purification

*Homo sapiens* or *Mus musculus* full-length *TMEM120A* (TACAN, residues 1–343) were cloned into pEG BacMam (*Goehring et al., 2014*). The C-terminus of the TACAN construct contains a PreScission protease cleavage site and an enhanced green fluorescent protein (eGFP) for purification (TACAN-eGFP). Briefly, bacmid carrying TACAN was generated by transforming *Escherichia coli* DH10Bac cells with the corresponding pEG BacMam construct according to the manufacturer's instructions (Bac-to-Bac; Invitrogen). Baculoviruses were produced by transfecting *Spodoptera frugiperda* Sf9 cells with the bacmid using Cellfectin II (Invitrogen). Baculoviruses, after two rounds of amplifications, were used for cell transduction. HEK293S GnTI- cells (ATCC, CRL-3022) grown in suspension at a density of ~3 × 10^6 cells/mL were transduced with P3 BacMam virus of TACAN-eGFP, and inoculated at 37 °C. 8–12 hr post-transduction, 10 mM sodium butyrate was added to the culture and cells were further inoculated for 40–48 hr at 30 °C. Cells were then harvested by centrifugation, frozen in liquid $N_2$, and stored at –80 °C until needed.

Frozen cells (from 1 L cell cultures) were resuspended in 200 mL hypotonic lysis buffer containing 50 mM Tris-HCl pH 8.0, 3 mM dithiothreitol (DTT), 1 mM ethylenediaminetetraacetic acid (EDTA), 0.1 mg/mL DNase I, and a protease inhibitor cocktail (1 mM PMSF, 0.1 mg/mL trypsin inhibitor, 1 µg/mL pepstatin, 1 µg/mL leupeptin, and 1 mM benzamidine) for 30 min and centrifuged at 37,500 g for 30 min. The pellets were then homogenized in 20 mM Tris-HCl pH 8.0, 300 mM NaCl, 0.1 mg/mL DNase I, a protease inhibitor cocktail followed by addition of 10 mM lauryl maltose neopentyl glycol (LMNG), 2 mM cholesteryl hemisuccinate (CHS) (for cryo-EM samples), or 1% n-decyl-β-D-maltopyranoside (DM), 0.2 % CHS (w/v) (for reconstition and mass spectrometry samples) to solubilize for 2 hr. The suspension was then centrifuged at 37,500 g for 30 min and the supernatant incubated with 5 mL GFP nanobody-coupled CNBr-activated Sepharose resin (GE Healthcare) for 2 hr (*Kubala et al., 2010*). The resin was subsequently washed with 10 column volumes of wash buffer containing 20 mM HEPES pH 7.4, 250 mM NaCl, and 0.06% digitonin (w/v) (for cryo-EM samples) or 0.25% DM, 0.05 % CHS (w/v) (for reconstition and mass spectrometry samples). The washed resin was incubated overnight with PreScission protease at a protein to protease ratio of 40:1 (w:w) to cleave off GFP and release the protein from the resin. The protein was eluted with wash buffer, concentrated using an Amicon Ultra centrifugal filter (MWCO 100 kDa), and then injected onto a Superdex 200 increase 10/300 GL column (GE Healthcare) equilibrated with the wash buffer. Peak fractions corresponding to the TACAN dimer were pooled. For cryo-EM study, the pooled fractions were concentrated to 6–7 mg/mL using an Amicon Ultra centrifugal filter (MWCO 100 kDa). All the purification steps were carried out at 4 °C.

*H. sapiens* full-length *ELOVL7* (residues 1–281) was cloned into the same vector, expressed, and purified with the same protocol as TACAN in DM/CHS. The final protein concentration was ~2 mg/mL.

## Proteoliposome reconstitution

Dialysis-mediated reconstitution of *H. sapiens* TACAN and ELOVL7 into liposome was accomplished according to published protocols with minor modifications (*Brohawn et al., 2012*; *Heginbotham et al., 1999*; *Long et al., 2007*; *Tao and MacKinnon, 2008*; *Wang et al., 2014*). Briefly, 20 mg of soy L-α-phosphatidylcholine (soy-PC) was dissolved in 1 mL chloroform in a glass vial and dried to a thin film under argon, rehydrated in reconstitution buffer (10 mM HEPES pH 7.4, 450 mM NaCl, and 2 mM DTT) to 20 mg/mL by rotating for 20 min at room temperature, followed by sonication with a bath sonicator until translucent. 1% DM was then added, and the lipid detergent mixture was rotated for 30 min and sonicated again until clear. Purified TACAN (~3 mg/mL) or ELOVL7 (~2 mg/mL) in DM/CHS and DM-solubilized lipids (20 mg/mL) were mixed at protein-to-lipid (w:w) ratios of 1:20, 1:50, and 1:100, incubated for 2 hr, and then dialyzed against 4 L reconstitution buffer for 4 days with daily exchange at 4 °C. Biobeads (Bio-Rad) were added to the reconstitution buffer for the last 12 hr. The resulting proteoliposomes were flash frozen in liquid $N_2$ and stored at −80 °C.

## GUV formation

The dehydration/rehydration-mediated blister formation technique was used for generation of GUVs as previously reported (*Brohawn et al., 2014*). In brief, an aliquot of reconstituted *H. sapiens* TACAN proteoliposome was thawed at room temperature and spotted onto the 14 mm glass coverslip inside a 35 mm glass-bottomed Petri dish (Mattek; P35G-1.5-14 C) as 4–6 similar-sized drops. Spotted proteoliposomes were then dried under vacuum at 4 °C for 6 hr followed by rehydration with ~20 µL rehydration buffer (10 mM HEPES pH 7.4, 140 mM KCl). The rehydration was done by sitting the 35 mm Petri dish inside a 15 cm Petri dish lined with wet filter paper overnight at 4 °C (~16 hr). 3 mL bath solution (10 mM HEPES pH 7.4, 140 mM KCl, 1 mM $MgCl_2$) was then added to the 35 mm dish before recording. Blisters were visible after ~10 min and were competent to form high-resistance seals for at least 2 hrs.

## Cell culture and transfection for patch recordings

CHO-K1 cells (ATCC) and piezo-1 knockout HEK-293T cells (established in this lab) were used for electrophysiology experiments because they have low endogenous mechanosensitive currents (*Brohawn et al., 2014*; *del Marmol, 2016*).

Cells were cultured in DMEM-F12 (Gibco) (CHO cells) or DMEM (Gibco) (HEK-293T cells) supplemented with 10% FBS, 2 mM L-glutamine, 100 units/mL penicillin, and 100 µg/mL streptomycin. Cells

were plated in 35 mm plastic dishes and grown to ~50–60% confluency at 37 °C. Right before transfection, culture media was replaced by DMEM-F12 or DMEM with 10% FBS and 2 mM L-glutamine. 1 µg of *H. sapiens* TACAN-eGFP or the M2 muscarinic receptor (Halo-M2R-eGFP) plasmid (previously established in this lab) was transfected into the cells using FugeneHD (Promega) following the manufacturer's protocol. Cells were transferred to 30 °C after transfection and recordings were carried out 16–18 hr post-transfection. Immediately before recording, media were replaced by the bath solution (10 mM HEPES pH 7.4, 140 mM KCl, 1 mM MgCl$_2$).

## Excised inside-out patch recordings

Pipettes of borosilicate glass (Sutter Instruments; BF150-86-10) were pulled to ~2–6 MΩ resistance with a micropipette puller (Sutter Instruments; P-97) and polished with a microforge (Narishige; MF-83). Recordings were obtained with an Axopatch 200B amplifier (Molecular Devices) using excised inside-out patch techniques. Recordings were filtered at 1 kHz and digitized at 10 kHz (Digidata 1440A; Molecular Devices). Pressure application through patch pipettes was performed with a high-speed pressure clamp (ALA Scientific) controlled through the Clampex software. Pressure application velocity was set to the maximum rate of 8.3 mmHg/ms. All recordings were performed at room temperature. Pipette and bath solutions were identical unless otherwise stated: 10 mM HEPES pH 7.4, 140 mM KCl, and 1 mM MgCl$_2$ (~300 Osm/L).

## Planar lipid bilayer recordings

The bilayer experiments were performed as previously described with minor modifications (*Ruta et al., 2003*; *Wang et al., 2014*). A piece of polyethylene terephthalate transparency film separated the two chambers of a polyoxymethylene block, filled with symmetrical buffer containing 10 mM HEPES pH 7.4, 150 mM KCl unless otherwise stated. A lipid mixture of DPhPC (1,2-diphytanoyl-sn-glycero-3-phosphocholine, Avanti, Cat# 850356):POPA (1-palmitoyl-2-oleoyl-sn-glycero-3-phosphate, Avanti, Cat# 840857) (3:1, w:w) dissolved in decane (20 mg/mL) was pre-painted over an ~100 µm hole on the transparency film. Voltage was controlled with an Axopatch 200B amplifier in whole-cell mode. The analog current signal was low-pass filtered at 1 kHz (Bessel) and digitized at 10 kHz with a Digidata 1550A digitizer (Molecular Devices). Digitized data were recorded on a computer using the software pClamp (Molecular Devices, Sunnyvale, CA). Experiments were performed at room temperature.

## Cell lines

All the cell lines except for Piezo1 knockout HEK-293T, which was previously generated in the lab, were purchased from ATCC, and we have confirmed there is no mycoplasma contamination for all of them.

## Cryo-EM sample preparation and data collection

For both the WT and His196Ala, His197Ala mutant of *M. musculus* TACAN, purified protein at a concentration of 6–7 mg/mL was mixed with 2.9 mM Fluorinated Fos-Choline-8 (FFC8; Anatrace) immediately prior to grid preparation. 3.5 µL of the mixture was applied onto a glow-discharged Quantifoil R1.2/1.3 400 mesh Au grid (Quantifoil), blotted for 4 s at room temperature (RT) with a blotting force of 2–4 and humidity of 100%, and plunge-frozen in liquid ethane using a Vitrobot Mark IV (FEI).

Cryo-EM data were collected on a 300-kV Titan Krios electron microscope (Thermo Fisher Scientific) equipped with a K2 Summit (TACAN[WT]), or a K3 Summit (TACAN[H196A H197A]) direct electron detector and a GIF Quantum energy filter set to a slit width of 20 eV. Images were automatically collected using SerialEM in super-resolution mode. After binning over 2 × 2 pixels, the calibrated pixel size was 1.03 Å with a preset defocus range from 0.7 to 2.1 µm (TACAN[WT]), or 0.515 Å with a preset defocus range from 0.8 to 2.2 µm (TACAN[H196A H197A]), respectively. Each image was acquired as either a 10 s movie stack of 50 frames with a dose rate of 7.54 e/Å$^2$/s, resulting in a total dose of about 75.4 e$^-$/Å$^2$ (TACAN[WT]), or a 1.5 s movie stack of 38 frames with a dose rate of 37.7 e/Å$^2$/s, resulting in a total dose of about 56.6 e/Å$^2$ (TACAN[H196A H197A]).

## Image processing

For TACAN$^{WT}$, the image processing workflow is illustrated in *Figure 3—figure supplement 1D*. A total of 2,071 super-resolution movie stacks were collected. Motion-correction, twofold binning to a pixel size of 1.03 Å, and dose weighting were performed using MotionCor2 (*Zheng et al., 2017*). Contrast transfer function (CTF) parameters were estimated with Gctf (*Zhang, 2016*). Micrographs with ice contamination were removed manually, resulting in 1,982 micrographs for further processing. A total of 583,766 particles were auto-picked using Relion 3.1 (*Scheres, 2020*; *Scheres, 2012*; *Zivanov et al., 2018*; *Zivanov et al., 2020*) and windowed into 256 × 256-pixel images. Reference-free 2D classification was performed to remove contaminants, yielding 383,719 particles. These particles were subjected to ab initio reconstruction in cryoSPARC-2.9.0 (*Punjani et al., 2017*), specifying four output classes. The best class with 245,031 particles was selected, then subjected to a resolution-based classification workflow similar to a previous study (*Kang et al., 2020*). In brief, 40 iterations of global search 3D classification (K = 1) in Relion 3.1 with an angular sampling step of 7.5° was performed to determine the initial alignment parameters using the initial model generated from cryoSPARC. For each of the last five iterations of the global search, a K = 6 multi-reference local angular search 3D classification was performed with an angular sampling step of 3.75° and a search range of 30°. The multi-reference models were generated using reconstruction at the last iteration from global search 3D classification low-pass filtered to 8, 15, 25, 35, 45, and 55 Å, respectively. The classes that showed obvious secondary structure features were selected and combined. Duplicated particles were removed, yielding 130,491 particles in total. These particles were subsequently subjected to non-uniform refinement with C2 symmetry in cryoSPARC, which resulted in a map with a resolution of 4.5 Å. Iterative cycles of non-uniform refinement in cryoSPARC with C2 symmetry and Bayesian polishing in Relion 3.1 with new training parameters were performed until no further improvement, resulting in a 3.7 Å map. The refined particles were further cleaned up with one round of ab initio reconstruction (K = 4) in cryoSPARC and 110,090 particles remained. Finally, these particles were subjected to the non-uniform refinement with C2 symmetry in cryoSPARC, which yielded the final map at 3.5 Å resolution.

For TACAN$^{H196A\ H197A}$, 10,541 super-resolution movie stacks were collected. Motion-correction, twofold binning to a pixel size of 0.515 Å, and dose weighting were performed using MotionCor2 (*Zheng et al., 2017*). CTF parameters were estimated with CTFFind4 (*Rohou and Grigorieff, 2015*). Micrographs with ice contamination were removed manually, resulting in 9,600 micrographs for further processing. A total of 1,474,917 particles were auto-picked using Relion 3.1 and windowed into 400 × 400-pixel images, then binned two times and subjected to 2D classification, yielding 975,636 particles. The following image processing workflow is identical to TACAN$^{WT}$ sample. Briefly, these particles were subjected to ab initio reconstruction in cryoSPARC-2.9.0 (*Punjani et al., 2017*), specifying four output classes. The best class with 607,159 particles was selected, then subjected to the resolution-based classification, yielding 391,137 particles. Subsequent non-uniform refinement with C2 symmetry in cryoSPARC was performed, resulting in a map with a resolution of 3.8 Å and the resolution was further improved to 3.3 Å by iterative Bayesian polishing and non-uniform refinement cycles. Particles were further cleaned up with one round of ab initio reconstruction with 155,946 particles remaining. Finally, these particles were subjected to the non-uniform refinement with C2 symmetry in cryoSPARC, which yielded the final map at 2.8 Å resolution.

The mask-corrected Fourier shell correlation (FSC) curves were calculated in cryoSPARC 2.9.0, and reported resolutions were based on the 0.143 criterion. Local resolutions of the final maps were estimated by Relion 3.1 (*Scheres, 2020*; *Zivanov et al., 2020*). A summary of reconstructions is shown in *Table 1* and *Figure 3—figure supplement 1E,F*, *Figure 3—figure supplement 2A,B*, *Figure 4—figure supplement 1B–E*.

## Model building and refinement

For TACAN$^{WT}$, the 3.5 Å resolution map was subjected to Buccaneer in the CCP-EM suite (*Burnley et al., 2017*; *Wood et al., 2015*) to generate the de novo model. This initial model was further improved using phenix.sequence_from_map in Phenix (*Adams et al., 2010*). Several iterative cycles of refinement using the phenix.real_space_refine with secondary structure and NCS restraints and manual adjustments in COOT yielded the final model for the TACAN$^{WT}$ containing residues 9–72, 76-250 and 262–335 (*Adams et al., 2010*; *Emsley et al., 2010*).

For TACAN[H196A H197A], model of TACAN[WT] was placed into the 2.8 Å map using UCSF Chimera (*Pettersen et al., 2004*) and manually adjusted in COOT *Emsley et al., 2010* followed by iterative refinement cycles using the phenix.real_space_refine in Phenix with secondary structure and NCS restraints and manual adjustments in COOT. The final model for TACAN[H196A H197A] contained residues 9–72, 76–250 and 262–335 as well as 2 CoASH molecules bound.

Refinement statistics are summarized in *Table 1*. Structural model validation was done using Phenix and MolProbity based on the FSC = 0.5 criterion (*Chen et al., 2010*). Figures were prepared using PyMOL (https://pymol.org/2/) and UCSF Chimera (*Pettersen et al., 2004*). Representative densities of TACAN[WT] and TACAN[H196A H197A] are shown in *Figure 3—figure supplement 2C* and *Figure 4—figure supplement 1F*, respectively.

## Native MS analysis

The purified wild-type and mutant *M. musculus* TACAN samples were incubated with excess (0.7–0.9 mM) ligand (CoASH, S-ethyl-CoA, or acetyl-CoA) for 1 hr on ice. The incubated samples were then buffer exchanged into 200 mM ammonium acetate, 0.002 % LMNG (2 CMC) using a Zeba microspin desalting column with a 40 kDa MWCO (Thermo Fisher Scientific). For nMS analysis, a 2–3 µL aliquot of each buffer-exchanged sample was loaded into a gold-coated quartz capillary tip that was prepared in-house and then electrosprayed into an Exactive Plus with extended mass range (EMR) instrument (Thermo Fisher Scientific) using a modified static direct infusion nanospray source (*Olinares and Chait, 2020*). The MS parameters used include spray voltage, 1.22–1.25 kV; capillary temperature, 125–200 °C; in-source dissociation, 125–150 V; S-lens RF level, 200; resolving power, 8,750 or 17,500 at m/z of 200; AGC target, $1 \times 10^6$; maximum injection time, 200 ms; number of microscans, 5; injection flatapole, 8 V; interflatapole, 7 V; bent flatapole, 5 V; high-energy collision dissociation (HCD), 200 V; ultrahigh vacuum pressure, $6–7 \times 10^{-10}$ mbar; and total number of scans, at least 100. The instrument was mass calibrated in positive EMR mode using cesium iodide.

For data processing, the acquired MS spectra were visualized using Thermo Xcalibur Qual Browser (v. 4.2.47). MS spectra deconvolution was performed either manually or with UniDec v. 4.2.0 (*Marty et al., 2015*; *Reid et al., 2018*). The UniDec parameters used included m/z range, 1500–5000; mass range, 20,000–100,000 Da; peak shape function, Gaussian; and smooth charge state distribution, on.

From their primary sequences, the expected masses for the proteins are TACAN[WT] monomer: 41,770 Da, TACAN[WT] dimer: 83,539 Da, TACAN[H196A H197A] monomer: 41,637 Da, and TACAN[H196A H197A] dimer: 83,275 Da. Experimental masses were determined as the average mass± standard deviation (SD) across all the calculated mass values in the relevant peak series (n ≥ 5 charge states) with typical SDs of ±1 Da.

## Enzymatic activity assay to measure coenzyme A release

Coenzyme A releasing activity was measured using a fluorescence-based coupled-enzyme assay (Sigma-Aldrich, Cat. MAK034) in a 96-well microplate (Costar) at 37 °C. The reaction was monitored with an Infinite-M1000 spectrofluorometer (Tecan) with 535 nm excitation and 587 nm emission. The reconstituted proteoliposomes of ELOVL7 and TACAN at 1:50 protein-to-lipid ratio were used (10 µg total protein), supplemented with 100 µM malonyl-CoA (Sigma-Aldrich, Cat. M4263) and 50 µM stearoyl-CoA (Sigma-Aldrich, Cat. S0802). Reaction mixtures were incubated at 37 °C for 0, 0.5, 1, 2, 4, 8, 24, and 48 hr, frozen in liquid N$_2$, and stored at –80 °C. The mixtures were then centrifuged at 20,817 g for 10 min, and supernatants were used to perform the enzymatic assay following the manufacturer's protocol.

## Acknowledgements

We thank Mark Ebrahim, Johanna Sotiris, and Honkit Ng at the Evelyn Gruss Lipper Cryo-EM Resource Center at Rockefeller University for assistance in data collection; Dr. Chia-Hsueh Lee (St. Jude Children's Research Hospital) for critical reading of the manuscript and suggestions for image analysis; Dr. Yixiao Zhang (Interdisciplinary Research Center on Biology and Chemistry) for advice and help on data collection; and members of the MacKinnon lab and Chen lab (Rockefeller University) for assistance. This work was supported in part by GM43949 (to RM) and GM109824 and GM103314 (to BTC). RM is an investigator in the Howard Hughes Medical Institute.

## Additional information

### Funding

| Funder | Grant reference number | Author |
|---|---|---|
| National Institutes of Health | GM43949 | Roderick MacKinnon |
| National Institutes of Health | GM109824 | Brian Chait |
| National Institutes of Health | GM103314 | Brian Chait |
| Howard Hughes Medical Institute | | Roderick MacKinnon |
| Rockefeller University | | Hanan Alwaseem |

The funders had no role in study design, data collection and interpretation, or the decision to submit the work for publication.

### Author contributions

Yiming Niu, Data curation, Formal analysis, Investigation, Methodology, Software, Writing – review and editing, Writing – review and editing; Xiao Tao, Data curation, Investigation, Methodology, Resources, Software, Writing – review and editing, Writing – review and editing; George Vaisey, Data curation, Investigation, Methodology, Software, Writing – review and editing, Writing – review and editing; Paul Dominic B Olinares, Data curation, Investigation, Methodology, Writing – review and editing; Hanan Alwaseem, Data curation, Formal analysis, Methodology, Writing – review and editing; Brian T Chait, Funding acquisition, Investigation, Project administration, Resources, Supervision, Writing – review and editing, Validation; Roderick MacKinnon, Funding acquisition, Project administration, Supervision, Validation, Writing – review and editing, Project administration, Investigation, Resources, Writing – review and editing

### Author ORCIDs

Yiming Niu http://orcid.org/0000-0002-5683-1781
Xiao Tao http://orcid.org/0000-0002-9381-7903
George Vaisey http://orcid.org/0000-0002-8359-1314
Paul Dominic B Olinares http://orcid.org/0000-0002-3429-6618
Hanan Alwaseem http://orcid.org/0000-0002-4946-1436
Roderick MacKinnon http://orcid.org/0000-0001-7605-4679

### Decision letter and Author response

Decision letter https://doi.org/10.7554/eLife.71188.sa1
Author response https://doi.org/10.7554/eLife.71188.sa2

## Additional files

### Supplementary files
• Transparent reporting form

### Data availability

The B-factor sharpened 3D cryo-EM density map and atomic coordinates of wild-type TACAN have been deposited in Worldwide Protein Data Bank (wwPDB) under accession codes 7N0K and EMD-24107. The B-factor sharpened 3D cryo-EM density map and atomic coordinates of His196Ala, His197Ala mutant TACAN have been deposited in Worldwide Protein Data Bank (wwPDB) under accession codes 7N0L and EMD-24108.

The following dataset was generated:

| Author(s) | Year | Dataset title | Dataset URL | Database and Identifier |
|---|---|---|---|---|
| Niu Y, Tao X, MacKinnon R | 2021 | Cryo-EM structure of TACAN in the apo form (TMEM120A) | https://www.rcsb.org/structure/unreleased/7N0K | RCSB Protein Data Bank, 7N0K |
| Niu Y, Tao X, MacKinnon R | 2021 | Cryo-EM structure of TACAN in the apo form (TMEM120A) | https://www.ebi.ac.uk/pdbe/entry/emdb/EMD-24107 | Electron Microscopy Data Bank, EMD-24107 |
| Niu Y, Tao X, MacKinnon R | 2021 | Cryo-EM structure of TACAN in the H196A H197A mutant form (TMEM120A) | https://www.rcsb.org/structure/unreleased/7N0L | RCSB Protein Data Bank, 7N0L |
| Niu Y, Tao X, MacKinnon R | 2021 | Cryo-EM structure of TACAN in the H196A H197A mutant form (TMEM120A) | https://www.ebi.ac.uk/pdbe/entry/emdb/EMD-24108 | Electron Microscopy Data Bank, EMD-24108 |

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
