## [Decision Letter]

**Acceptance summary:**

Tmem120a/TACAN was recently identified as a potential mechanosensitive ion channel that mediates mechanical pain. Here, the authors use electrophysiology and structural biology to show that Tmem120a/TACAN does not display mechanically activated currents and does not structurally resemble an ion channel. Instead, this work shows that Tmem120a has structurally homology to a fatty acid elongase, ELOVL thus providing clues to this molecule's true physiological function.

**Decision letter after peer review:**

Thank you for submitting your article "Analysis of the Mechanosensor Channel Functionality of TACAN" for consideration by *eLife*. Your article has been reviewed by 3 peer reviewers, one of whom is a member of our Board of Reviewing Editors, and the evaluation has been overseen by Kenton Swartz as the Senior Editor. The following individual involved in the review of your submission as agreed to reveal their identity: Sudha Chakrapani (Reviewer #3).

The reviewers have discussed their reviews with one another, and the Reviewing Editor has drafted this to help you prepare a revised submission. Overall, this is a timely and well-executed study that convincing shows that Tacan/Tmem120a is not a mechanosensitive ion channel. The reviewers reached the consensus that the following minor revisions are needed before publication:

1) Please provide an expanded introduction (see comments by reviewer #1).

2) Please provide more details/quantification of the electrophysiology data (see detailed comments by reviewer #2).

3) Additionally, if you have the data, all reviewers felt seeing other examples of heterogeneous conductance produced when different (non-ion channel) membrane molecules are reconstituted? This would support the general idea of that recordings reflect non-specific disruptions of the lipids by high concentrations of membrane proteins.

4) Please explain the rationale behind the His mutations a bit more (see comment by reviewer #3).

5) Although not required, please consider expanding the discussion to include any thoughts about the physiological role of Tacan/Tmem120a in nociceptors (see comment by reviewer #1).

*Reviewer #1:*

Tmem120a was initially identified as a nuclear localized enzyme important for adipocyte differentiation and lipogenesis. More recently this molecule was proposed to function as a mechanically-gate ion channel important for the transduction of mechanical forces in nociceptors. That study provides two key pieces of evidence to support their conclusions: that expression of Tmem120a was sufficient to endow cells with mechanically gated currents and hat this function could be reconstitution cell-free in proteo-liposomes. These data are challenged in the current study from Niu and colleagues. They are unable reproduce these findings and instead propose the high concentrations of Tmem120a cause heterogeneous membrane disruptions that do not resemble any known type of ion channel gated. More importantly, they have solved the cryo-EM structure which convincing shows that Tmem120a is not an ion channel but instead has homology to a fatty acid elongase, ELOVL. Other interesting features of the structure include the binding pocket for coenzyme-A co-factor that may be helpful to future work focused on better elucidated the physiological function of this molecule. Overall, the manuscript is very straight forward and convincing. Future work is need to elucidate the functional role(s) Tmem120a may play in pain sensation and other physiological processes.

Specific comments to the authors:

Because I am not an expert in cryo-EM, I am focusing my comments on the background, context and electro-physiological data:

The authors need to expand the introduction, although this is a short report, more context is needed as to what was previously known/unknown about Tmem120a. Perhaps, it would be helpful to include some information about the other molecules proposed to be involved in mechano-sensation and perhaps the criteria previously used to support various claims.

It would have been nice to include some positive control data from a bonafide mechanosensitive ion channel, if the authors already have some.

Do other transmembrane proteins (that are not ion channels) also cause non-selective conductances when reconstituted?

Although it is clear that Tacan is not an ion channel, the authors have not addressed the in vitro or behavioral aspects of the Beaulieu-Laroche study. Is it possible that the enzymatic activity of Tmem120a could indirectly effect mechano-transduction?

*Reviewer #2:*

A recent study had identified TACAN (TMEM120A) as a putative mechanosensitive channel with important physiological implications. In order to better understand the mechanisms underlying the mechanosensitivity of TACAN, the authors set out to functionally and structurally characterize this ion channel. Surprisingly, they were unable to detect mechanically activated currents in cellular and reconstituted contexts. The authors also used cryo-electron microscopy to determine a structure of TACAN. Rather than resembling an ion channel, the protein shared structural homology to a fatty acid elongase (ELOVL7). The authors found density for coenzyme-A in the homologous catalytic cleft and confirmed the identity of the ligand using mass spectrometry.

The results in this study clearly support the authors' claim that TACAN is not a mechanosensitive ion channel. Structural biology is a particularly useful tool in this case as the structural similarity to ELOVL7 and the presence of a similar coenzyme in the catalytic site are striking. The functional data appear to be convincing as well; however, only representative traces are shown, and more rigorous quantification of the data is necessary.

Overall, this study is of great importance to the field. The discovery and molecular characterization of bonafide mechanosensitive channels is of high interest due to the relative lack of identified proteins. This report succinctly and convincingly demonstrates that this proposed candidate is likely not actually a mechanically activated channel.

As presented, the manuscript concisely makes a strong case that TACAN is not actually a mechanosensitive channel. As stated in the public review, this kind of study is very important for the field to corroborate the findings of a high-profile publication. Although there are a few experiments that could potentially strengthen the study, these are likely not necessary for the main point of the paper. However, prior to acceptance of this paper in *eLife*, quantitative analyses of the electrophysiology data are necessary.

– In figure 1, it would be good to show open probability values of all traces in the presence and absence of pressure to convincingly show that there is no significant change in activation in response to mechanical stimuli.

– The representative traces in figure 2 do suggest that currents are heterogeneous in amplitude as stated in the text; however, amplitude histograms for single channel data would be helpful to further illustrate this point.

– IV curves in figures 2C and 2F appear to be based on n of 1. If this is the case, a higher n is necessary. It might also be helpful to clearly state in the text that the shift in reversal potential indicates that the larger NMDG cation is less permeable through the proposed "leaky membrane".

– In figure panels 2A and 2D, it would be helpful to indicate where in the trace "closed" is indicated, since it appears that some of the traces have inward and outward currents.

Other experiments to consider, but are not necessary for acceptance:

– Reconstituting other membrane proteins that are known to not be ion channels (such as the M2R control used in HEK and CHO) into GUVs at similarly high concentrations that caused leaks in TACAN would help determine whether the leaky membrane effect is TACAN-specific.

– The orientation of the protein with respect to the cytoplasm (line 62) could be determined with a tagging approach in cells.

*Reviewer #3:*

The manuscript by Niu et al., reports the functional properties and molecular architecture of TMEM120A/TACAN. TMEM120A has generated enormous interest due to its potentially divergent functional roles. It was originally identified as a transmembrane nuclear envelope protein expressed in adipocytes, and involved in fat metabolism. More recently, renamed as TACAN, TMEM120A was reported to be a mechanically-activated ion channel expressed in a subset of nociceptor neurons, and a potential therapeutic target in the management of chronic pain. The latter study showed slowly adapting, stretch-induced currents in excised patches from cells heterologously expressing TACAN as well as from reconstituted liposomes containing purified TACAN. Some intriguing behavior of TACAN were that there was no response to poking-stimulation in whole-cell configuration and the currents had markedly different single-channel conductance in cells vs liposomes.

In this work, the authors carried out electrophysiological recordings under similar conditions as in the previous study. Contrary to the previous findings, here they did not observe any stretch-induced currents in excised patches from cells (CHO and HEK) expressing TACAN or in patches from reconstituted proteoliposomes. Under conditions of very high protein-to-lipid ratio, highly heterogenous single-channel behavior were observed that were non-selective to permeant ions and unresponsive to mechanical stimuli. The authors conclude that these currents are likely due to membrane leak artifacts. Although in the previous study it was reported that currents (including those from liposomes) were blocked by Gd3+ and GsMTx4 toxin.

The authors solved cryo-EM structures of TACAN WT and a double mutant (to improve the ligand density) which revealed a dimeric architecture, with each monomer consisting of 6 TMs and two N-terminal helices that form a coiled-coil. Overall, the architecture bears close resemblance to ELOVL7, a member of the long chain fatty acid elongase family. With elaborate native mass-spec analysis, they establish the identity of the co-purified ligand (coenzyme-A) and its stoichiometry.

Overall, this is a well-executed study and a timely report.

---

## [Author Response]

The reviewers have discussed their reviews with one another, and the Reviewing Editor has drafted this to help you prepare a revised submission. Overall, this is a timely and well-executed study that convincing shows that Tacan/Tmem120a is not a mechanosensitive ion channel. The reviewers reached the consensus that the following minor revisions are needed before publication:1) Please provide an expanded introduction (see comments by reviewer #1).

We have expanded the Introduction to provide additional background. We also revised the Discussion to address an issue raised in the review. In our opinion the electrophysiological experiments provide the primary evidence that TACAN is not an MSC and the structural experiments lend support to that conclusion. In the revised discussion we explain this view.

2) Please provide more details/quantification of the electrophysiology data (see detailed comments by reviewer #2).

The absence of a discernable response to patch pressurization is so clear that open probability graphs would look strange in our opinion. Papers often publish processed versions of data with insufficient primary data. Here we show primary data. For the same reason we do not think that amplitude histograms, Fourier transforms or any other such analysis will add to the obvious and only conclusion we are trying to make, that we do not see mechanically activated currents.

We removed the I-V curves in Figure 2 because they are unnecessary. We have carried these out many times and thought it worth showing an example of a conductance (not mechanosensitive) that exhibits almost no selectivity (weak selectivity for a metal cation over a larger organic cation). Now we realize this will serve a distraction, especially if we provide a table with statistics on this weak selectivity. Those who use techniques of ion channel reconstitution will recognize the conductance behavior we show here as “junk channels”.

On page 3 (lines 60-62) we added a sentence on TRAAK channels in GUVs using the same methods.

3) Additionally, if you have the data, all reviewers felt seeing other examples of heterogeneous conductance produced when different (non-ion channel) membrane molecules are reconstituted? This would support the general idea of that recordings reflect non-specific disruptions of the lipids by high concentrations of membrane proteins.

We do not have data on whether other proteins produce heterogeneous conductance in reconstituted membranes. We do not know whether they do.

4) Please explain the rationale behind the His mutations a bit more (see comment by reviewer #3).

A sentence was added to page 4 (lines 94-95) explaining the rationale behind the histidine to alanine mutations.

5) Although not required, please consider expanding the discussion to include any thoughts about the physiological role of Tacan/Tmem120a in nociceptors (see comment by reviewer #1).

It is possible that TACAN/Tmem120a affects nociceptor function. We did not study this.